# Factors Influencing Emergency Department Nurses’ Compliance with Standard Precautions Using Multilevel Analysis

**DOI:** 10.3390/ijerph18116149

**Published:** 2021-06-07

**Authors:** Su Jung Kim, Eun Ju Lee

**Affiliations:** 1Dongsan Medical Center, Division of Nursing, Keimyung University, Daegu 42601, Korea; sjsy283@naver.com; 2Department of Nursing, Keimyung University, Daegu 42601, Korea

**Keywords:** emergency department, multilevel analysis, precautions, standards of care

## Abstract

Standard precautions protect patients and nurses from infection. Nevertheless, compliance with standard precautions is lower among emergency department nurses than other nurses. We examined the individual and organizational factors that influence emergency department nurses’ compliance with standard precautions via a cross-sectional study. A self-reported questionnaire survey was administered to 140 nurses working in nine emergency departments in South Korea. It included items regarding ethical awareness and standard precaution self-efficacy at the individual level as well as safety environment, organizational culture for infection control, and degree of compliance with standard precautions at the organizational level. Individual and organizational predictors were identified using a multilevel analysis. The results indicated that 81.1% of nurses’ compliance with standard precautions was influenced by individual differences, while only 18.9% was influenced by organizational differences. Individual- and organizational-level predictors explained 46.7% and 55.4% of the variance in emergency department nurses’ compliance with standard precautions, respectively. Emergency department nurses’ compliance with standard precautions was predicted by ethical awareness and standard precaution self-efficacy at the individual level and by organizational culture for infection control at the organizational level. Our findings provide evidence for the need to improve facilities and human resource management as well as the organizational culture for infection control.

## 1. Introduction

Standard precautions (SPs), published by the Center for Disease Control and Prevention (CDC) [1], apply to all environments and cases where healthcare workers contact patients. It is a key component of healthcare-associated infection control for the primary prevention of the spread of bloodborne and other pathogens. Other components include hand hygiene, appropriate use of personal protective equipment (PPE), safe use and disposal of injection needles, elimination of environmental and equipment contamination, patient placement, and management of textiles and wastes.

The Healthcare Infection Control Practice Advisory Committee [2] emphasized that assuring avoidance of exposure to potential infection sources when handling blood and bodily fluids is important to ensure the safety of patients and health professionals and should be practiced for all patients. By complying with SPs, nurses can prevent exposure to potential infection sources, thus ensuring both their patients’ safety and their own [3]. The spread of the Middle East Respiratory Syndrome (MERS) provided an opportunity to promote hospital-acquired infection management and national epidemic prevention and management, which included ameliorating the healthcare accreditation system and improving the emergency healthcare system alongside promoting compliance with SPs among healthcare professionals [4]. Since then, the outbreak of the novel coronavirus (COVID-19), which spreads directly through droplets and indirectly through aerosols, has highlighted the need for rigorous infection control among healthcare professionals [5,6].

### Background

An emergency department (ED) is a complex and dynamic healthcare environment characterized by patients with infectious diseases, such as hepatitis A, tuberculosis, MERS, and COVID-19, who are waiting to be diagnosed [4]. It has a higher infection rate than patient wards or intensive care units (ICUs) [7] because 70% of the nursing activities taking place increase the likelihood of exposure to blood or bodily fluids, thereby endangering the safety of healthcare professionals and patients [8]. Therefore, EDs are the places of first response to prevent not only hospital-acquired infections but also the spread of community-acquired infections. Accordingly, aggressive and systematic infection control is crucial [4].

Nevertheless, compliance with SPs is lower in EDs than in outpatient clinics, patient wards, and ICUs owing to frequent emergency situations and a shortage of organizational facilities and supplies [8,9]. Compliance with SPs among ED nurses is influenced not only by nurses’ individual characteristics, but also by the organizational environment [10]. This can be described using the ecological systems theory, which posits that humans interact with their environment and cannot be isolated from it [11]. Thus, when examining the factors that influence nurses’ care practices, the organizational environment surrounding nurses must also be considered [12,13].

Previous studies have not investigated the impact of organizational factors on compliance with SPs [9,10,14,15,16,17]. However, healthcare professionals’ compliance with SPs varies according to the features of their organizations, such as facilities for hand hygiene, staffing, and organizational culture for infection control [10]. As patients’ severity, established culture, and environment also differ according to the type of emergency healthcare facility, the predictors of compliance with SPs should be analyzed considering various organizational factors [10,18]. However, few studies have both individual and organizational levels in their analyses [13,19]. In this context, it is necessary to identify the organizational factors that reflect the features of EDs and the individual characteristics of ED nurses.

Previous studies on internal factors related to compliance with SPs identified general characteristics such as education level [15] and specific characteristics such as SP self-efficacy [20], knowledge, and attitude [15]. However, SP self-efficacy, knowledge, and attitude were only confirmed to be correlated, while education level and awareness were identified as predictors. In addition, these studies were limited in shedding light on the factors influencing nurses at the individual level. Nurses’ ethical awareness is correlated with patient safety management activities [21] and adherence to nursing practice guidelines [22]. In particular, SPs are highly likely to be overlooked in the ED owing to time-pressing and emergency situations [8], which call for good ethical awareness among ED nurses. Thus, it is important to examine the impact of ethical awareness among ED nurses on their compliance with SPs.

Factors in organizational environments that influence compliance with SPs include organizational culture for infection control, size, type of ED [10,23], and safety environment [9,12,13,24,25,26,27,28,29,30]. In particular, the differences in facilities, equipment, and human resources affect not only the nurse-to-patient ratio in emergency healthcare facilities but also the formation of an organizational culture for infection control and safety environment [10]. The safety environment refers to the personnel and physical environment needed to comply with SPs [16]; compliance with SPs is related to accessibility to PPE and the amount of time available [17]. Furthermore, although the nurse-to-patient ratio is a predictor of nurses’ compliance with SPs [13], none of the previous studies were conducted in the ED. Thus, it is necessary to examine the level of compliance with SPs considering the structure of the ED, nurse-to-patient ratio, safety environment, and organizational culture for infection control.

Statistical analytical techniques, e.g., regression analysis, that have been used in prior studies can only reveal the influence of individual or organizational variables on the dependent variable and cannot shed light on inter-level interactions [27,28,29,30]. A multilevel analysis, on the other hand, is beneficial in confirming the influence of organizational factors on compliance with SPs [29,31]. In other words, among nurses who share the same work environment and culture, both individual- and organizational-level variables must be considered on analyzing factors of compliance with SPs [13]. Thus, the objective of this study was to identify the factors influencing ED nurses’ compliance with SPs using multilevel analysis by categorizing them into individual factors, including ethical awareness and self-efficacy, and organizational factors, including emergency room features, nurse-to-patient ratio, safety environment, and organizational culture for infection control.

## 2. Materials and Methods

### 2.1. Design and Participants

A cross-sectional study design was used. In addition, this study was conducted according to the STROBE checklist.

Participants in this study were nurses who worked in the ED for more than two months and performed direct nursing in D and Y city. The reason for more than two months of working experience in the ED was that this time was considered to allow 4–6 weeks of orientation after placement at the ED.

The sample size was determined using G*Power, version 3.1.9. For multiple regression analyses with a significance level of 0.05, effect size of 0.15, power of 0.80, and 12 predicting variables, including general characteristics, infection-control-related characteristics, and individual and organizational factors, the minimum required sample size was 127. Considering a 10% withdrawal rate, the questionnaire was distributed to 140 participants. A total of 140 questionnaires were obtained. After an initial review of the collected questionnaires, those with incorrect responses were returned to the participant for re-completion and were collected again.

### 2.2. Instruments

#### 2.2.1. SP Self-Efficacy

SP self-efficacy refers to an individual’s belief or expectation of successfully performing tasks related to SP, which is a key element of healthcare-related infection control [32]. This was developed especially for nurses, with higher scores indicating a stronger sense of self-efficacy associated with SP. SP self-efficacy was measured using the SP self-efficacy instrument developed by Mohammed et al. [17], after obtaining permission from the authors. This tool consists of seven items, with each item rated on a four-point Likert scale. The reliability of the tool, as measured with Cronbach’s α, was 0.73 at the time of development [16] and 0.81 in this study.

#### 2.2.2. Ethical Awareness

Ethical awareness refers to an individual or collective view or ideas of ethics regarding the behavioral norm that a person must follow [33]. The higher the score, the higher the level of ethical awareness. This was measured using the ethical awareness scale developed by Jang [34] and modified based on the Korean Code of Ethics for Nurses by Youk et al. [35]. This scale comprises seven items, with each item rated on a four-point Likert scale and with reverse scoring for negatively worded items. The higher the score, the higher the level of ethical awareness. The reliability of the tool, as measured with Cronbach’s α, was 0.71 at the time of development [34], 0.71 in the study by Youk et al. [35], and 0.80 in this study.

#### 2.2.3. Safety Environment

The safety environment refers to a physical work and human resource environment that is necessary for nurses to comply with SP. This was measured using a tool related to SPs, which was developed by Cho [36], modified by Suh and Oh [16], and modified by H. J. Park [18] with reference to ED features and infection control SPs presented by the Korean Disease Control and Prevention Agency (KDCA). The tool comprises nine items, with “yes” or “no” responses for each item. The total score ranges from 0 to 9. The higher the score, the safer the environment. Cronbach’s α for the tool was not presented at the time of development; however, it was 0.70 in the study by H. J. Park [18] and 0.77 in this study.

#### 2.2.4. Organizational Culture for Infection Control

Organizational culture for infection control refers to the shared organizational culture that encompasses values, beliefs, customs, and norms recognized by organizations and shared by members in relation to healthcare-related infection control guidelines [37]. This was measured using the Hospital Survey on Patient Safety Culture developed by the Agency for Healthcare Research and Quality [37], translated by Kim et al. [38], modified by Park [39], and adapted to create a tool for organizational culture for infection control by Moon and Jang [40], in compliance with hospital-acquired infection management guidelines. This tool comprises 10 items, with each item rated on a seven-point Likert scale. The higher the score, the higher the level of awareness of organizational culture for infection control. The reliability of the tool, as measured with Cronbach’s α, was 0.78 at the time of development [37], 0.78 in the study by Park [39], 0.85 in the study by Moon and Jang [40], and 0.87 in this study.

#### 2.2.5. Compliance with SPs

Compliance with SPs refers to the degree to which SPs recommended by the CDC are implemented to prevent the spread of pathogens, including bloodborne infectious agents, regardless of the presence of active infection [17]. This was measured using a tool revised by the CDC [1], translated by Jung [41], modified by excluding infection control during lumbar puncture and patient placement by Hong et al. [42], and modified and adapted for use by hospital nurses by Lee [43]. This tool comprises 36 items, with each item rated on a five-point Likert scale. The higher the score, the higher the compliance with the SPs. The reliability of the tool, as measured with Cronbach’s α, was 0.95, in the study by Hong et al. [40], 0.96 in the study by Lee [43], and 0.93 in this study.

### 2.3. Data Collection

Data were collected from 140 ED nurses working in nine hospitals among the 32 hospitals with emergency rooms in cities D and Y of the Republic of Korea. Although the request for data collection was sent to 32 hospitals, only nine hospitals approved the data collection. Of all the ED nurses in the nine hospitals, a total of 140 nurses met the inclusion criteria. The researcher met all 140 nurses in person to explain the purpose and process of the study, and all 140 nurses agreed to participate in the study. The organizations were classified into three types based on the facility, equipment, and staffing standards: regional emergency medical centers, regional emergency healthcare institutions, and regional emergency healthcare facilities. Data were collected between December 2019 and February 2020. Prior to data collection, we obtained permission from the manager over the phone and visited the nursing departments at each organization to explain the purpose of the study to the teaching director and ED head nurse and request permission and cooperation for data collection. The researchers visited an hour before the start of each shift to explain the purpose of the study and the data collection procedure to the subjects and received consent forms from all participants. In addition, they collected questionnaires 20 min after distribution and ensured that there were no missing data by reviewing the questionnaires to identify any insufficient data. As a result, we were able to include all ED nurses who worked at the facility without any withdrawals.

### 2.4. Statistical Analysis

The collected data were analyzed using SPSS software (version 23.0, IBM Corporation, Daegu, Korea) and the R software lme4 package. Participants’ general and infection-control-related characteristics and other study parameters were analyzed using numbers, percentages, means, and standard deviations. Differences in general and infection-related characteristics and compliance with SPs were analyzed using t-tests and analyses of variance, followed by Scheffé’s post hoc test. The relationships between SP self-efficacy, ethical awareness, safety environment, organizational culture for infection control, and compliance with SPs were analyzed using Pearson’s correlation coefficients. Individual and organizational predictors were analyzed using a multilevel analysis.

### 2.5. Ethical Considerations

The study was conducted in accordance with the guidelines of the Declaration of Helsinki and approved by the Institutional Review Board of Keimyung University (no. 40525-201908-HR-044-01, date of approval: 11/12/2019). Written informed consent was obtained from those who voluntarily wished to participate. Participants were guaranteed that personal information would be coded to ensure anonymity and that the retrieved questionnaires would be stored in a locked cabinet for three years, after which they would be disposed of using a paper shredder.

## 3. Results

### 3.1. Differences in Compliance with SPs According to Participants’ General Characteristics, Infection Control-Related Characteristics, and Organizational Characteristics

Most participants were women in their 20s, had a bachelor’s degree, and had less than five years of work experience. Regarding infection-control-related characteristics, most had received education about infection control, and many had been wounded by a needle or sharp object in the past year and/or had their mucosa (eyes, mouth) or skin with an open wound exposed to patients’ blood or bodily fluids in the past year. Compliance with SPs differed significantly based on the history of a cut or puncture injury in the past year (t = 1.45, *p* = 0.022). There were no significant differences in compliance with SPs based on the participants’ general and organizational characteristics (Table 1).

### 3.2. Level of SP-Related Factors

Nurses’ mean SP self-efficacy score was 2.96 (±0.54) out of 4, and their mean ethical awareness score was 3.09 (±0.26) out of 4. The mean safety environment score was 6.92 (±1.60) out of 9, and the mean organizational culture for infection control score was 5.26 (±0.83) out of 7. The mean compliance with SP scores among ED nurses was 4.29 (±0.49) out of 5 (Table 2).

### 3.3. Correlation of SPs with Relevant Factors

Compliance with SPs was positively correlated with self-efficacy (r = 0.64, *p* < 0.001), ethical awareness (r = 0.46, *p* < 0.001), safety environment (r = 0.34, *p* < 0.001), and organizational culture for infection control (r = 0.54, *p* < 0.001) (see Table 3).

### 3.4. Multilevel Analysis of the Predictors of Compliance with SPs

In the base model analysis, the intraclass correlation coefficient (ICC) was 0.189, and the deviance was 201.91. An ICC value of 0.05 indicates suitability for multilevel analysis; thus, the study model was suitable for the analysis [28,40]. We used grand-mean centering, which is generally used for multilevel analyses.

In Model 1, the organizational model, the estimated fixed effect was 2.40 (*p* < 0.001). Safety environment (*p* = 0.041) and organizational culture for infection control (*p* < 0.001) significantly influenced compliance with SPs (*p* < 0.001). The residual variance was 0.687, indicating that 68.7% of the total variance was explained by organizational variables. Model 1 had a goodness of fit with a deviation of 146.25, which was reduced from 201.91 in the base model.

In Model 2, the individual model, the estimated fixed effect was 1.87 (*p* < 0.001). SP self-efficacy (*p* < 0.001) and ethical awareness (*p* = 0.007) were both significant. Regarding random effects, the individual-level variance decreased from 0.240 in the base model to 0.109, and the organizational-level variance decreased from 0.056 to 0.033. A greater reduction in organizational-level variance suggests that organizational variables (e.g., infection control organizational culture with a significant impact on tissue variables) explained a portion of the variation between organizations. The individual variables explained 54.6% of the variance, showing that they explained inter-individual differences. The coefficient of determination (R^2^) for Model 2 was 46%. The deviation was 201.91 for the base model and 110.61 for Model 2, showing that Model 2 had a better fit than Model 1.

In Model 3, which included both organizational and individual variables, the estimated fixed effect was 1.23. Significant predictors were SP self-efficacy (*p* < 0.001) and ethical awareness (*p* = 0.026) at the individual level and organizational culture for infection control (*p* < 0.001) at the organizational level. Regarding random effects, individual-level variance was 0.128 and organizational-level variance was 0.025, both of which were significant (*p* < 0.001). The individual-level variance decreased from 0.240 in the base model to 0.128, and the organizational-level variance decreased from 0.056 to 0.025, confirming the good fit of this model. The residual organizational variance was 0.546. After adding variables to the basic model, the organizational variables had an effect of approximately 68.7%, whereas the effect of the individual variables was 54.6%. Additionally, the addition of individual- and organizational-level variables showed that individual-level variables for compliance with SP in ED nurses had an effect of 46.7%, while organizational-level variables had an effect of 55.4%. The R^2^ value of Model 3 was 53%. The deviations were 201.91 for the base model, 146.25 for Model 1, 110.61 for Model 2, and 89.33 for Model 3, showing that Model 3 had the best fit (Table 4).

## 4. Discussion

This study examined compliance with SPs among ED nurses in South Korea. We used multilevel analysis to identify the predictors of compliance with SPs based on the interactions between individual and organizational factors.

Compliance with SPs among ED nurses varied by 18.9%, according to the characteristics of their organizations. The intergroup difference in compliance with SPs was 27.1% among hemodialysis nurses in the study by Kim and Shin [13], which was higher than that found in our study. This reflects the fact that EDs are more rigorously managed by laws and regulations than hemodialysis units. Therefore, strict national legal regulations, monitoring, and organizational management are crucial.

Organizational features explained 68.7% of the variance in compliance with SPs among ED nurses. When individual characteristics were considered, the explained variance decreased by 13.3%, to 55.4%. Individual factors accounted for 54.6% of the variance. When organizational factors were considered, the explained variance decreased by 7.9%, to 46.7%. The reason that the degree of influence was reduced when organizational and individual factors were considered together can be attributed to the interaction between individual characteristics and the features of their organizations. These results support the ecological systems theory, which argues that both individual factors and the environment surrounding an individual must be considered when analyzing factors related to human behavior [11].

The significant predictors of ED nurses’ compliance with SPs were organizational culture for infection control at the organizational level and SP self-efficacy and ethical awareness at the individual level. The fact that the influence of these predictors decreased when considered together as opposed to when individual and organizational predictors were considered separately shows that these predictors interact in their influence on nurses’ compliance with SPs. This suggests that even when individual nurses have low ethical awareness of self-efficacy, a good organizational culture for infection control can boost their compliance with SPs.

The mean compliance with SPs among ED nurses was 4.29 out of 5, which was similar to the compliance of 4.31 found in a study on ED nurses by Kim and Park [10], which used the same instrument. Han et al. [9] reported a score of 4.78 in outpatient services and 4.5 in medical and surgical wards, which were higher than those among ED nurses (4.09). The greater severity and number of patients being presented to the ED compared to other units and the consequent workload seems to lower compliance with SPs among ED nurses.

Organizational culture for infection control, a significant organizational predictor, was also a significant predictor in previous studies that performed a single-level analysis [10,40], thereby supporting our findings. Cumbler et al. [44] reported that individual compliance with SPs increased by 22.4% after fostering a positive organizational culture by implementing feedback for infection control, promoting sharing of responsibility with colleagues, and implementing a managerial reward or punishment system for infection control behaviors. In addition, Newstrom and Davis [45] emphasized that a positive organizational culture is an organizational factor that alters individual behaviors. Thus, organizational effort is needed to provide continuous feedback, implement communication training, and improve teamwork as a facilitator to alter individual behaviors to establish an organizational culture for infection control.

The individual predictors of ED nurses’ compliance with SPs were SP self-efficacy and ethical awareness. The mean SP self-efficacy score was 2.96. This is similar to previous results, in which compliance with SPs increases with an increase in SP self-efficacy [20]. However, the previous study only examined a correlation, and it was difficult to compare our results with existing findings owing to a lack of studies that analyze SP self-efficacy as a predictor. Most previous studies measured general self-efficacy as a predictor of compliance with SPs [10,40,46,47,48,49,50]; however, general self-efficacy did not reflect self-efficacy during work [49]. Thus, this study is significant in using SP self-efficacy and specifically measuring nurses’ confidence in complying with SPs. Subsequent studies should further investigate the impact of SP self-efficacy on ED nurses’ compliance with SPs.

Although no previous study has examined the relationship between ethical awareness and compliance with SPs among ED nurses, our results are similar to previous findings that showed that compliance with patient safety activities, including infection control, increases with increased ethical awareness [21] and that ethical awareness is a predictor of nurses’ infection control behaviors [22]. As shown here, ethical awareness influenced nurses’ infection control behaviors, and there is a need for replication studies on compliance with SPs and ethical awareness as well as additional studies that examine the individual-level moderating variables that facilitate compliance with SPs in this relationship.

Taken together, compliance with SPs among ED nurses was influenced by an interaction between a positive organizational culture for infection control and individuals’ self-efficacy and ethical awareness. Furthermore, organizational culture for infection control had a great influence on compliance with SPs among ED nurses (55.4%), highlighting the need to implement a feedback system and invest effort to improve communication and teamwork to foster a positive organizational culture for infection control. Therefore, to promote compliance with SPs, organizations need to implement education programs for SP self-efficacy and ethical awareness and employ aggressive policy interventions targeting organizational factors such as organizational culture for infection control.

The data for this study were collected from two cities by convenience sampling, thus entailing a regional and institutional bias as the study limitation. However, the results of this study are significant in identifying the factors influencing SPs by selecting nine hospitals with various types of emergency rooms and reflecting the unique characteristics of various facilities. In addition, it was difficult to compare our findings with the prior literature owing to a lack of studies that examine nurses’ SP self-efficacy, ethical awareness, and compliance with SPs; thus, replication studies may prove beneficial.

## 5. Conclusions

This cross-sectional study identified the predictors of compliance with SPs among ED nurses using multilevel analysis. The individual factors (SP self-efficacy and ethical awareness) and organizational factors (safety environment and organizational culture for infection control) were identified as the predictors of nurses’ compliance with SPs. In the multilevel analysis, the base model analysis confirmed that 18.9% of compliance with SPs among ED nurses was explained by differences among organizations. When both individual- and organizational-level variables were entered, the individual-level variables explained 46.7% of the variance, while organizational-level variables explained 55.4% of the variance.

### Relevance to Clinical Practice

To promote compliance with SPs among ED nurses, it is important to (i) foster an organizational culture that facilitates compliance with SPs, and (ii) implement measures to cultivate SP self-efficacy and ethical awareness among individual nurses. Thus, intervention programs are needed to improve SP self-efficacy, ethical awareness, and organizational support. Continuous management is needed to foster an organizational culture for infection control. This study serves as the basis for improving the organizational management of facilities and human resources as well as the organizational culture for infection control. The findings can be used as foundational data for developing interventions that boost compliance with SPs among ED nurses.

## Figures and Tables

**Table 1 ijerph-18-06149-t001:** Differences in compliance with SPs according to general characteristics, infection-control-related characteristics, and organizational features (N = 140).

Characteristic		Category	Frequency	Percentage	SPs Compliance		
					M ± SD	t or F	*p*
Individual level	Sex	Female	116	82.9	4.27 ± 0.05	0.91	0.864
		Male	24	17.1	4.36 ± 0.10		
	Age (years)	≤29	101	72.1	4.27 ± 0.48	1.25	0.294
		30–39	25	17.9	4.42 ± 0.09		
		40–49	14	5.7	4.08 ± 0.18		
		≥50	6	4.3	4.28 ± 0.04		
		Total			29.30 ± 0.61		
	Education level	Associate	17	12.1	4.27 ± 0.14	0.26	0.855
		Bachelors	111	79.3	4.30 ± 0.06		
		Masters	10	7.1	4.18 ± 0.17		
		Doctoral	2	1.4	4.47 ± 0.04		
	Total career (years)	<5	91	65.0	4.32 ± 0.05	0.65	0.586
		5–9	24	17.1	4.21 ± 0.09		
		10–14	11	7.9	4.30 ± 0.12		
		≥15	14	10.0	4.17 ± 0.18		
		Total			5.65 ± 0.59		
	ED career (years)	<1	38	27.1	4.29 ± 0.08	0.09	0.965
		1–2	51	36.4	4.30 ± 0.07		
		3–4	23	16.4	4.24 ± 0.10		
		≥5	28	20.2	4.29 ± 0.09		
		Total			3.25 ± 0.30		
	Infection control education	Yes	102	73.9	4.32 ± 0.05	1.24	0.986
		No	38	27.1	4.20 ± 0.08		
	Cut/puncture injuries	Yes	58	41.4	4.35 ± 0.05	1.45	0.022
		No	82	58.6	4.24 ± 0.06		
	Number of cut/puncture injuries (n) ^1^	1	28	48.3	4.31 ± 0.09	0.87	0.456
		2	13	22.4	4.52 ± 0.11		
		≥3	17	29.3	4.39 ± 0.08		
	History of exposure to blood and bodily fluids	Yes	43	30.7	4.31 ± 0.07	0.47	0.823
		No	97	69.3	4.27 ± 0.05		
	Number of exposures to blood and bodily fluids (n) ^2^	1	9-	20.9	4.30 ± 0.15	0.19	0.901
		2	11	25.6	4.38 ± 0.14		
		≥3	23	53.5	4.31 ± 0.11		
Organizational level	Type of ED	Regional emergency medical center	75 (2)	53.6 (22.2)	4.23 ± 0.47	1.16	0.317
		Regional emergency healthcare institution	40 (4)	28.6 (44.4)	4.35 ± 0.50		
		Regional emergency healthcare facility	24 (3)	17.9 (33.3)	4.35 ± 0.53		

	Number of patients per nurse	≤5	31	22.1	4.45 ± 0.48	2.26	0.108
		6–9	20	14.3	4.27 ± 0.49		
		≥10	89	63.6	4.23 ± 0.49		

^1^ Only includes those with a history of cut/puncture injury; ^2^ Only includes those with a history of exposure to blood and bodily fluids. Abbreviations: ED, emergency department; SPs, standard precautions.

**Table 2 ijerph-18-06149-t002:** Levels of factors related to standard precautions (N = 140).

Variable	Range	Mean ± SD	Min.	Max.
SP self-efficacy	1–4	2.96 ± 0.54	1.29	4.00
Ethical awareness	1–4	3.09 ± 0.26	2.50	4.00
Safety environment	0–9	6.92 ± 1.60	1.00	9.00
Organizational culture for infection control	1–7	5.26 ± 0.83	3.00	7.00
Compliance with SPs	1–5	4.29 ± 0.49	2.94	5.00

Abbreviation: SP, standard precaution.

**Table 3 ijerph-18-06149-t003:** Correlations among factors related to SPs (N = 140).

	SP Self-Efficacy	Ethical Awareness	Safety Environment	Organizational Culture for Infection Control	Compliance with SPs
	r (*p*)	r (*p*)	r (*p*)	r (*p*)	r (*p*)
SP self-efficacy	1				
Ethical awareness	0.48 (<0.001)	1			
Safety environment	33 (<0.001)	0.35 (<0.001)	1		
Organizational culture for infection control	0.47 (<0.001)	0.42 (<0.001)	0.39 (<0.001)	1	
Compliance with SPs	0.64 (<0.001)	0.46 (<0.001)	0.34 (<0.001)	0.54 (<0.001)	1

Abbreviation: SP, standard precaution.

**Table 4 ijerph-18-06149-t004:** Multilevel analysis of the predictors of compliance with SPs.

Fixed Effect												
	Base Model			Model 1			Model 2			Model 3		
Intercept (*ϒ*₀₀)	Coef.	SE	*p*	Coef.	SE	*p*	Coef.	SE	*p*	Coef.	SE	*p*
	4.29	0.04	<0.001	2.40	0.27	<0.001	1.87	0.37	<0.001	1.23	0.42	<0.001
**Level 1: individual (n = 140)**				t	*p*		t	*p*		t	*p*	
ED career							0.77	0.443		0.74	0.460	
Infection education: yes							1.23	0.223		0.56	0.550	
Cut/puncture injury: yes							0.99	0.325		1.03	0.302	
Exposure to blood and bodily fluids: yes							−0.52	0.603		−0.14	0.809	
SP self-efficacy							7.02	<0.001		6.26	<0.001	
Ethical awareness							2.75	0.007		2.25	0.026	
**Level 2: organizational (n = 9)**												
Type of ED				0.19	0.846					1.76	0.081	
Number of patients per nurse				0.23	0.818					0.24	0.806	
Safety environment				2.07	0.041					0.64	0.526	
Organizational culture for infection control				6.50	<0.001					3.91	<0.001	
**Random Effect**												
	**Base Model**			**Model 1**			**Model** **2**			**Model 3**		
Individual-level variance	0.240	<0.001		0.075	<0.001		0.109	<0.001		0.128	<0.001	
Organizational-level variance	0.056			0.017			0.033			0.025		
Individual-level residual variance							0.546			0.467		
Organizational-level residual variance				0.687						0.554		
Deviation	201.91			146.25			110.61			89.33		
R^2^	0.189 (ICC)			0.31			0.46			0.53		

Abbreviations: SE, standard error; Coef., coefficient; ICC, intraclass correlation coefficient; SPs, standard precautions; ED, emergency department; Model 1, organizational level; Model 2, individual level; Model 3, individual plus organizational level.

## Data Availability

Data are available on request due to privacy restrictions and ethical concerns. The data presented in this study are available upon request from the corresponding author. The data are not publicly available as the IRB requires material to be disposed of within three years.

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
