# Peer review of "Factors Influencing Emergency Department Nurses’ Compliance with Standard Precautions Using Multilevel Analysis"

_ijerph, 2021, doi:10.3390/ijerph18116149_

Round 1
Reviewer 1 Report
It is my pleasure to review this interesting study. The theme is novel and interesting. This research provides a scope to understand compliance with SPs among ED nurses. I had some suggestions to improve the paper.
What is definition of variables? And what is the relationship between the independent variable and the dependent variable?
What does the score measured using each tool mean?
Please provide the reference of the reliability at the time of development.
Reviewer 2 Report
The study addresses an important current issue for public health, emphasizing ED professionals. However, it has limitations in a few important aspects.
Introduction: It is generally well described. It addresses the problem, justify the study and describe the objectives. Nevertheless, objetives are too "ambitious", considering the data avaiable.
Methods: The authors mention they followed the STROBE checklist. However, they did not clearly describe how they selected the 140 participants. Was it a simple random selection? Stratification by institution was considered? The authors mention later that "EDs in Y city were additionally conveniently sampled". More information on how the participants were selected and recruited would be necessary. What reasons justify the collection of data in only in these workplaces? Also, it is not clear the number of workers that represents the target population and how many people were from each place.
Regarding the tools used to measure the main variables, were they validated for the target population? Was the tool used to measure the outcome in other studies? Why did the authors showed results in means? What is the clinical relevance of these 'means'? What scores (or cutoffs) could be classified as acceptable or adequate?
Results: Some mistakes in tables and p values would need corrections.
Table 1: Mean column is not necessary.
Table 4 presentation is confusing. Models are not clear. Did the authors include any sociodemographic factor in the models. How and why the variables were selected to be included?
Discussion: Important limitations of the study are discussed, impacting negatively in the generalization of the findings.
The authors concluded that "This study serves as the basis for improving organizational management of facilities and human resources as well as the organizational culture for infection control.". However, generalization of the findings are limited. Inferences might be done to a very particular group in Korea. The sample was not selected at random. Authors would need to improve discussion, and reinforce weaknesses and limitations in their paper. Based on this, strong conclusions would not be possible, so they should be softened.
There are several other papers on this subject - including reviews - that could be considered for this study.
Examples:
Wong EL, Ho KF, Dong D, Cheung AW, Yau PS, Chan EY, Yeoh EK, Chien WT, Chen FY, Poon S, Zhang Q, Wong SY. Compliance with Standard Precautions and Its Relationship with Views on Infection Control and Prevention Policy among Healthcare Workers during COVID-19 Pandemic. Int J Environ Res Public Health. 2021 Mar 25;18(7):3420. doi: 10.3390/ijerph18073420.
Vaismoradi, M.; Tella, S.; A. Logan, P.; Khakurel, J.; Vizcaya-Moreno, F. Nurses’ Adherence to Patient Safety Principles: A Systematic Review. Int. J. Environ. Res. Public Health 2020, 17, 2028. https://doi.org/10.3390/ijerph17062028
Oh, J.; Cho, H.; Kim, Y.Y.;Yoo, S.Y. Validation of the Korean Version of the Nursing Profession Self-Efficacy Scale: A Methodological Study. Int. J. Environ. Res. PublicHealth 2021, 18, 1080. https://doi.org/10.3390/ijerph18031080
Reviewer 3 Report
In general, the manuscript is well organized, structured and presents clear results according to the proposed methodology.
However, some aspects can be improved, in the part of the introduction from line 106 to line 113 it would go in the methodology section in "statistical analysis".
Next, in the "Instrument" section, it should come in plural and is not singular, since more than one instrument has been passed (line 131).
The statistical analyzes and models are correct. The discussion is clear and comparing with other good quality studies. Finally, the conclusion responds to the objective of the study.
Reviewer 4 Report
First of all, congratulations on your work. It was a pleasure to read.
Background
The background is clear and logical. The research question is a direct translation of the problem that arose from the background.
Material and Methods
The methodology is clear and leads the readers to an understanding of the statistical options. Table 1 could influence the selection of the potential predictors. However, introducing all the variables in the model is also justifiable.
No ethical issue appears to arise from the study. Guiding the research upon the Declaration of Helsinki guidelines was a good option.
Results
Table 1 - what do you intend to present with "b < a2".
Discussion
The discussion evidences an adequate analysis of the results well contextualized and compared with other similar studies.
Conclusion
The authors are able to translate the results and discussion and summarize them in interesting findings for practice implementation.
Overall, a pleasure to read. Congratulations once again for your hard work.
Round 2
Reviewer 2 Report
Major revisions.
"Despite its interesting and important subject, several comments and suggestions were not revised in the manuscript.
Objectives are still confusing. Authors repeated the same sentence concerning the main objective (lines 104-106) and added some information (lines 107-110) that should be placed only in methods section. The authors also added a new sentence (lines 110-113) that is not actually adequate for an introduction. The relevence of the study and its implication for policies should be discussed based on the findings (discussion section).
Selection of participants is still not clear. The reader is not able to know what the target population is. How many nurses in total worked in each ED of all 9 institutions selected for the study? What criteria was used to select institutions and individuals? What was the response rate? Authors did not clarify these aspects. If it is a convenience sample, it should be specified and discussed as a limitation. This issue was also not addressed in the discussion section.
Regarding the measurements, the clinical relevance of scores is still not clear. What scores (or cutoffs) could be classified as acceptable or adequate for each factor authors evaluated? How were they calculated? The outcome (compliance with SP) was measured using a 36-item likert scale. However, mean scores of compliance with SP varied from 4.08 to 4.52. In the methods section, authors only mention that "the higher the scores, the best", but they did not specify their meanings.
Table 4 is still confusing and shows unnecessary information.
Considering these aspects, discussion should still be improved."
Author Response
"Please see the attachment."
